# Calcineurin-Dependent Homeostatic Response of *C. elegans* Muscle Cells upon Prolonged Activation of Acetylcholine Receptors

**DOI:** 10.3390/cells12172201

**Published:** 2023-09-03

**Authors:** Franklin Florin, Benjamin Bonneau, Luis Briseño-Roa, Jean-Louis Bessereau, Maëlle Jospin

**Affiliations:** 1Institut NeuroMyoGène, CNRS UMR-5284, INSERM U-1314, MeLiS, Université Lyon, Université Claude Bernard Lyon 1, F-69008 Lyon, Francebenjamin.bonneau@universite-paris-saclay.fr (B.B.); jean-louis.bessereau@univ-lyon1.fr (J.-L.B.); 2Institut Curie, CNRS UMR3347, INSERM U1021, Université Paris-Saclay, F-91405 Orsay, France; 3Medetia Pharmaceuticals, Institut Imagine, F-75015 Paris, France

**Keywords:** calcium, muscle, homeostasis, calcineurin, acetylcholine receptor, *C. elegans*

## Abstract

Pharmacological adaptation is a common phenomenon observed during prolonged drug exposure and often leads to drug resistance. Understanding the cellular events involved in adaptation could provide new strategies to circumvent this resistance issue. We used the nematode *Caenorhabditis elegans* to analyze the adaptation to levamisole, an ionotropic acetylcholine receptor agonist, used for decades to treat nematode parasitic infections. Genetic screens in *C. elegans* identified “adapting mutants” that initially paralyze upon exposure to levamisole as the wild type (WT), but recover locomotion after a few hours whereas WT remain paralyzed. Here, we show that levamisole induces a sustained increase in cytosolic calcium concentration in the muscle cells of adapting mutants, lasting several hours and preceding a decrease in levamisole-sensitive acetylcholine receptors (L-AChR) at the muscle plasma membrane. This decrease correlated with a drop in calcium concentration, a relaxation of the animal’s body and a resumption of locomotion. The decrease in calcium and L-AChR content depends on calcineurin activation in muscle cells. We also showed that levamisole adaptation triggers homeostatic mechanisms in muscle cells including mitochondria remodeling, lysosomal tubulation and an increase in autophagic activity. Levamisole adaptation thus provides a new experimental paradigm for studying how cells cope with calcium stress.

## 1. Introduction

Prolonged exposure to a pharmaceutical component often results in adaptation to that substance and, consequently, a reduction in its efficacy. For example, long-term exposure to nicotine leads to changes in the amount and activity of acetylcholine receptors, contributing to the addiction phenomenon. The molecular characteristics of adaptation are not always well understood. Levamisole is a small chemical compound that efficiently activates ionotropic acetylcholine receptors in nematodes, while it is poorly efficient on mammalian nicotinic receptors. It has therefore been used extensively as a potent anthelminthic since the 1960s [1] to treat devastating intestinal nematode parasite infections in cattle, and has also been used in humans. Resistance to levamisole has developed over time: for example, a nationwide survey in Ireland found that levamisole treatment was only about 50% effective in sheep [2]. A better understanding of the cellular events induced by levamisole could provide new strategies to circumvent this resistance issue.

Studies on *Ascaris suum* have established that levamisole is an agonist of ionotropic acetylcholine receptors (AChR): it acts on muscle cells and triggers depolarization of the plasma membrane [3,4]. Subsequent studies, carried out in parasitic species and the free-living nematode *Caenorhabditis elegans*, have shown that levamisole activates some AChR (L-AChR) present at the surface of muscle cells [5,6,7]. Genetic screens for mutants surviving high concentrations of levamisole identified key-components of nicotinic neurotransmission in *C. elegans* [8,9,10]. Mutants totally insensitive to levamisole lack L-AChRs but are viable and motile because a second class of levamisole-insensitive receptors is present at the neuromuscular junctions (NMJs). These screens enabled the identification of genes encoding L-AChR subunits [11,12,13] and proteins essential for receptor biosynthesis or trafficking [14]. A second class of mutants initially paralyze on levamisole but recover locomotion after several hours of exposure, a phenomenon called adaptation [9]. In these mutants, L-AChR are expressed but are overall less functional as compared to WT because regulatory mechanisms are altered [15,16,17,18,19,20]. Among these regulatory proteins, MOLO-1 is an auxiliary subunit that promotes the gating of receptors [20], CRLD-1 enhances the assembly of receptors in the endoplasmic reticulum (ER) [18]. A group of three proteins, LEV-9, LEV-10 and OIG-4 interact to form an extracellular scaffold required for L-AChR clustering at NMJs [15,16,17].

Why disrupting the synaptic clustering of L-AChRs would cause adaptation to levamisole remained unexplained for several years. Indeed, initial in vivo electrophysiological experiments showed that levamisole induced similar currents when applied on muscle cells of wild type and in L-AChR-clustering-defective mutants [15,16,17]. Hence, levamisole should be as efficient in these mutants as in WT. In a recent study [21], however, we have shown that the clustering machinery modifies the biophysical properties of the receptors. Levamisole-elicited currents are indeed similar in muscle cells of wild type and clustering mutants when recorded at −60 mV, as previously described [15,16,17], but they are halved in these mutants at −30 mV, a value close to the physiological resting potential of nematode muscle cells [22,23,24]. This phenomenon was explained by the fact that L-AChR clustering confers a strong outward rectification of the current, which is absent when clustering proteins were mutated [21]. As a result, in wild type, levamisole induces a strong depolarization that causes the full inactivation of voltage-gated calcium channels. These channels physiologically control the excitability of the muscle cells and mediate the rising phase of action potentials in *C. elegans* [22,23,24]. In L-AChR-clustering-defective mutants, levamisole-induced depolarization is lower than in WT, which is not sufficient to induce the full inactivation of voltage-gated calcium channels [21]. Calcium flows into the cells through these channels and calcium levels remain high for a minimum of 4 h. It remains unclear how muscle cells of adapting worms are able to survive the continuous entry of calcium into muscle cells. It is even more puzzling that the animals are able to recover their movement while still being exposed to the depolarizing drug.

In this study, we have investigated the cellular mechanisms underlying adaptation to levamisole. We determined that cytosolic calcium concentration decreases in muscle cells of adapting mutants, after its initial sustained rise, within a time window similar to that observed for locomotion recovery. We showed that this phenomenon is correlated with a decrease in L-AChR content at the muscle plasma membrane and we provided evidence that the calcium-dependent phosphatase calcineurin is involved in these processes. Finally, we showed that the changes in cytosolic calcium concentration observed during levamisole adaptation cause dramatic remodeling of cellular organelles with modifications of mitochondria, lysosomes and autophagic vesicles.

## 2. Materials and Methods

### 2.1. Strains and Genetics

All *C. elegans* strains were originally derived from the wild-type Bristol N2 strain. Worm cultivation, genetic crosses, and manipulation of *C. elegans* were carried out according to standard protocols [8]. All strains were maintained on nematode growth medium (NGM) agar plates with *Escherichia coli* OP50 as a food source at 20 °C. Null mutants are referred to as *mut-x(0)*. A complete list of strains used in this study can be found in Appendix A.

### 2.2. Behavioral Assays

Levamisole was dissolved in water at a concentration of 1 M and kept at −20 °C. A few days before performing assays, levamisole was added to 55 °C-equilibrated NGM medium at the final concentration of 1 mM. Levamisole-containing plates were then seeded with OP50. One-day old adult hermaphrodites were transferred on fresh NGM plates for control condition or on levamisole plates, and plates were stored at 20 °C. Paralysis after levamisole treatment was scored at room temperature (19–22 °C). Each experiment was performed on at least 3 different days.

Speed and length measurements were performed in brightfield, using a Nikon Multizoom AZ100. Speed was measured for 50 s at 2 frames per second. Worms’ centroids were tracked using the TrackMate plugin in Fiji [25]. For worms with multiple tracks, only the first track was selected. Tracks of less than 10 s were excluded. Length was measured from head to tail of one-day adults with a custom Fiji macro, and corresponded to the distance approximately separating the pharynx procorpus and the anus, following the center of the worm lateral section.

### 2.3. Calcium Imaging

For calcium imaging, animals carried a single copy of *Pmyo-3::GCaMP-6s-mCherry::unc-54 3′UTR*, inserted by mini-Mos [26] using a plasmid derived from pJH3630[*Pmyo-3::GCaMP-6s-mCherry::unc-54 3′UTR*] (kindly gifted from Mei Zhen, Samuel Lunenfeld Research Institute), and the *lite-1(ce314)* mutation to limit light reactivity during imaging [27]. Worms were transferred on thick (about 0.5 cm) agarose pads placed on glass slides. Agarose pads were composed in mass of 1.6% agarose, 0.3% NaCl, 1% 5 mg/mL cholesterol in ethanol, completed with deionized water. 1 mM levamisole and/or 1 mM auxin were added before pouring when mentioned. 3 worms were placed per pad, with a thin OP50 layer. During experiments, glass slides were kept in a humid atmosphere to prevent drying.

Imaging was performed with an Eclipse Ti2 Nikon microscope, equipped with a Chroma 59022x excitation filter and Chroma 59022bs dichroic mirror, and illuminated with a pE-300^white^ LED system. Emission was split using an OptoSplit II system equipped with Chroma T565lpxr dichroic mirror and Chroma ET525/50m and ET632/60m emission filters. Both color channels were imaged side by side with a Prime BSI Scientific CMOS camera. Imaging was performed in situ, without coverslip. Two images per second were taken for a minimum of 30 s and up to 45 s with a 20/0.75 objective. Mobile worms were followed using a custom tracker repositioning the stage near the worm vulva using Python programming language, Pymmcore [28], or using a convolutional artificial neural network for worm center recognition using the TensorFlow [29] library. Muscle signal was collected from a strip area following the worm outline. Background signal was determined by selecting areas outside of the worm. Images were processed using a custom Fiji macro.

The ratio for an image was calculated as follows:$$R=\frac{1}{N_{ROI}}\sum_{i\in ROI}\frac{I_{GCaMP}(i)}{I_{mCherry}(i)}$$
with *R* being the mean ratio, used to quantify a single image, *ROI* the set of pixels of interest, *N_ROI_* the number of *ROI* elements, *I_Channel_ (i)* the intensity value of the pixel *i* for a given channel after subtraction of the mean background value of the channel.

The plotted normalized ratio value of a full recording was calculating as follows:$$\frac{∆R}{{∆R}_{max}}=\frac{\frac{1}{N_{T}}\sum_{i\in T}R_{l}-R_{min}}{R_{max}-R_{min}}$$
with *T* being the set of images for an individual worm’s full recording, *N_T_* the number of images, *R_max_* the maximum image mean ratio recorded in the first 90 s of levamisole exposure (*R_max_* = 3.12) and *R_min_* the minimum image mean ratio recorded in the exposure of worms to 10 mM muscimole (*R_min_* = 0.504).

### 2.4. Modifications of the Endogenous Loci

To generate the *kr423* and *kr540* alleles, the Auxin-Inducible Domain (AID) encoding sequence [30] and respectively *mNeonGreen(mNG)* or *eBFP* sequences were inserted tandemly into the *tax-6* locus, just before the stop codon, using CRISPR/Cas9 technology [31]. CrRNA sequences targeting the insertion region were designed and synthesized from IDT (Integrated DNA Technologies Inc., Coralville, IA, USA). CrRNA and tracerRNA (trRNA) were mixed as 1:1 ratio to form a sgRNA duplex. Adult worms were micro-injected in the gonads with a mix containing Cas9 nuclease 0.5 μL of 10 μg/μL (IDT Inc., Coralville, IA, USA), sgRNA duplex 3 μL (100 μM), repair templates (containing the DNA fragment to insert, a hygromycin resistance selection cassette surrounded by *loxP* sites, *Phsp16.41::Cre* and homology arms to target the genomic region) 50 ng/μL, RNase/DNase free water up to 10 μL. Injected animals were then grown at 25 °C, and after 48 h, a positive selection was performed by adding hygromycin on plates at 0.2 μg/μL. Resistant animals were then heat-shocked for 2 h at 34 °C to excise the hygromycin selection cassette from the genome. Worms were then screened based on fluorescence signal in muscle cells.

Engineering of *crh-1(bab180[crh-1-aid])* and *crtc-1(kr450[crtc-1-aid-wScarlet])* were done as described above. *aid* or *aid-wScarlet* sequences were, respectively, inserted at the start of exon 2 of *crh-1* or just before the stop codon of *crtc-1*. Animals with successfully modified genome were screened by PCR for *bab180* and by tracking fluorescence signal in muscle cells for *kr450*.

### 2.5. Auxin-Induced Degradation

Auxin plates were prepared by adding auxin indole-3-acetic acid from a 400 mM stock solution in ethanol into NGM at the final concentration of 1 mM. Tissue-specific degradation was achieved by expressing TIR-1 element fused to eBFP under *eft-3*, *rab-3*, *myo-3* or *dpy-7* promoters. For experiments with the different tissue-specific rescues, animals were transferred on auxin plates from L4 stage. For all other experiments, animals have hatched on auxin plates.

### 2.6. Microscopy Imaging and Quantification

For all microscopy experiments, live one-day adults were mounted on 2% agarose dry pads immersed in 5% poly-lysine beads diluted in M9 buffer (3 g of KH_2_PO_4_, 6 g of Na_2_HPO_4_, 5 g of NaCl and 0.25 g of MgSO_4_·7 H_2_O, distilled water up to 1 L).

Mitochondria imaging was performed on animals expressing a single copy of *Pmyo-3::tom-20-wScarlett* [32]. Mitochondria morphology was rated blindly using the aforementioned Eclipse Ti2 microscope from 7 to 20 muscle cells per animal. Each cell was rated as exhibiting either fragmented, elongated or networking mitochondria. The most frequent value was used for each worm.

Muscle autophagic activity was monitored from animals expressing the integrated *Pdyc-1::gfp-lgg-1* or *Pdyc-1::gfp-lgg-1(G116A)* transgenes as previously described [33,34]. GFP puncta were visualized using the aforementioned Eclipse Ti2 microscope and were manually counted from 5 to 20 cells per animal, in blinded conditions. A mean number was then calculated for each animal. Autophagic vesicles were also observed in animals expressing the integrated *Plgg-1::mCherry-GFP-lgg-1* transgene, which allows a dual staining of the autophagosomes and lysosomes [35]. In these animals, autophagosomes were counted using a custom Fiji macro only in cell parts that were distant from the epidermis.

For ligand-gated receptor quantification, images were acquired using a confocal Andor spinning disk system (Oxford Instruments) installed on an Olympus IX83 microscope equipped with a 40/1.3 and a 60/1.4 oil-immersion objectives and an Evolve electron-multiplying charge-coupled device camera. Light excitation was performed with laser centered at 561 nm wavelengths, while emission went through a Semrock FF02-617/73-25 filter. Images were acquired by IQ 3.4.1 software from Andor. Dorsal cord fluorescence quantification was performed as previously described [36], with total summed depth varying for each cord acquisition, and with a quantification crop size of 40 µm (wide) × 3 µm (high). Unless specified, normalized values plotted are the fluorescence values divided by the mean of the corresponding control condition of the same genotype without levamisole.

Muscle ATP was evaluated using the Queen-2m probe as previously described [37]. Images were acquired using the aforementioned confocal spinning disk microscope with the 40/1.3 objective. Light excitation was performed with lasers centered at 405 nm and 488 nm wavelengths, respectively, while emission went through a Semrock FF03-525/50-25 filter. A ratio was obtained using the same calculation process as with calcium imaging applied on the sum of 0.3 µm distant planes taken over the lens-exposed half of the worm. For each worm, three stacks were acquired, on the head and at respectively the upmost and bottommost locations of the anterior and posterior gonads. The plotted value corresponds to the average ratio between the three areas.

Muscle lysosomal morphology was visualized in animals expressing a single copy of *Pmyo-3::lmp-1-eBFP*, inserted by mini-Mos using a plasmid carrying *Pmyo-3::lmp-1-eBFP*. 3D images of the lens-exposed half of the worm were taken between nerve ring and vulva using the aforementioned confocal microscope and rated blindly as exhibiting circular, tubular or networking lysosomes using a custom Fiji macro. Lysosomal length was estimated using a custom Fiji macro using the particle’s skeleton total length. Individual particle’s length was plotted for 1637 to 3356 particles per condition (14 to 16 worms per condition, two to five 3D images per worm).

### 2.7. Electrophysiology

Microdissection of *C. elegans* and in vivo electrophysiology were performed as previously described [38]. The pipette solution contained (in mM) 120 KCl, 4 NaCl, 5 EGTA, 10 TES, 4 MgATP, sucrose to 300 mosm/L (pH 7.2), and, the bath solution contained (in mM) 150 NaCl, 5 KCl, 1 CaCl_2_, 1 MgCl_2_, 10 glucose, 15 HEPES, 0.01 dihydro-beta-erythroidine (DHbE) and sucrose to 310 mosm/L (pH 7.2). The open cut preparation was constantly exposed to the bath solution by placing them in the mouth of a perfusion tube from which flowed by gravity the rapidly exchanged solutions. Levamisole at a concentration of 0.01 mM was present in the bath solution all along the experiments. For voltage-clamp experiments, holding membrane potential was −60 mV. Membrane currents were measured in response to voltage ramps from −60 to 0 mV, then cells were exposed to 0.1 mM of d-tubocurarine chloride (dTBC) and currents were measured again. For each cell, currents measured in the presence of dTBC were subtracted from those in the absence of dTBC to obtain the dTBC-sensitive levamisole current. All experiments were performed at 20 °C.

### 2.8. Statistical Analysis

For all violin plots, dotted lines show quartiles, dashed lines denote mean and tests compare levamisole condition to the control of the same mutant. All error bars represent standard deviations from the mean. For all p values: *** *p* < 0.0005, ** *p* < 0.005, * *p* < 0.05, ns (not significant) *p* ≥ 0.05. Statistical analyses were performed using GraphPad Prism (GraphPad Software, Boston, MA, USA, https://www.graphpad.com/).

## 3. Results

### 3.1. Movement Recovery in Adapting Mutants Is Associated with Muscle Relaxation

When placed on 1 mM levamisole plates, WT animals paralyze and eventually die after several hours. Animals with partially defective L-AChRs, as *lev-10* or *molo-1* mutants, become initially paralyzed, but then recover locomotion after 6 to 8 h of drug exposure, a phenomenon called adaptation to levamisole (Figure 1a). Although almost 100% of the mutant animals move on the plate, measuring the speed of the adapted animals showed that locomotion is slow as compared to non-treated animals (Figure 1b). Notably, the relative restoration of speed on plates was higher for *lev-10(0)* worms than for *molo-1(0)*.

We then followed muscle contraction over time using as a proxy the length of the worms in WT and *lev-10(0)* mutants (Figure 1c). Upon levamisole exposure, *lev-10(0)* and WT worms initially contract to the same extent. While *lev-10(0)* remained contracted after 15 min, WT started to relax. At 4 h, *lev-10(0)* reached a maximal contraction, with a length of 70% compared to initial values, while WT were at 92%. After 10 h, when movement was already recovered, *lev-10(0)* adapting worms had progressively relaxed up to 85% of their base length.

Besides locomotion, we also monitored the effect of levamisole on food intake. *C. elegans* feeding relies on rhythmic contraction of the pharynx, which was shown to be impacted by levamisole [39]. As previously described [39], we observed that the number of pumping events was inhibited after 5 min on levamisole in WT and in *lev-10(0)* mutant (Appendix A). After 2 h of levamisole exposure, feeding remains inhibited in WT while it started to recover in *lev-10(0)* mutant. After 24 h, the number of pumping events returned to 86% of the initial value in *lev-10(0)* mutant while it was still very low in WT.

### 3.2. Movement Recovery in Adapting Mutants Is Associated with a Restoration of Muscle Calcium Levels

In a previous study [21], we showed that the initial contraction induced by levamisole was accompanied by a rise in calcium levels which persists in *lev-10(0)* adapting worms for up to 4 h. The following relaxation and movement restoration observed in these adapting animals could be caused by either a decrease in intracellular calcium after 4 h, or a decrease in calcium sensitivity of the contractile apparatus. To distinguish among these two hypotheses, we monitored muscle cytosolic calcium concentration over time during levamisole exposure.

We built animal strains expressing a single copy insertion of the pseudo-ratiometric indicator GCaMP-6s::mCherry in striated muscles, using a *lite-1(0)* genetic background [27] to minimize contractions induced by the blue light stimulation used for GCaMP excitation. These animals exhibited a homogeneous expression of the fluorophore in muscle cells and did not display any visible developmental or locomotion defect. We measured average calcium concentration on freely moving WT and adapting animals after 2 and 10 h of levamisole exposure (Figure 1d). In the absence of levamisole, calcium levels in muscle were similar in WT, *lev-10(0)* and *molo-1(0)*. At 2 h, calcium was significantly higher in adapting mutants, in agreement with previous results obtained by different researchers. After 10 h on levamisole, *lev-10(0)* muscle calcium was reduced below the levels of WT worms. In *molo-1(0),* calcium levels were more heterogeneous among the population with some animals showing strong calcium reduction while others continued exhibiting high calcium concentrations and remaining hyper-contracted. This might explain why the movement speed recovery of the *molo-1(0)* strain is not as efficient at 10 h as compared to *lev-10* mutants (Figure 1b).

Altogether, these results showed that movement recovery of adapting worms was correlated with a decrease in calcium levels in muscle cells after an initial increase.

### 3.3. Movement Recovery in Adapting Mutants Is Associated with a Decrease of Levamisole-Sensitive Receptor Content

We proposed the hypothesis that the decrease in muscle calcium level observed in adapting strain after adaptation could result from a downregulation of L-AChR. Using a knock-in strain expressing a fluorescently-tagged version of UNC-29, an obligatory L-AChR subunit, we measured the quantity of synaptic L-AChR before and after adaptation in the WT and in three adapting strains, *lev-10(0)*, *molo-1(0)* and *crld-1(0)* (Figure 2a,b and Appendix A). Although the basal level of synaptic receptors differed in each strain, we saw an approximate 30% decrease in fluorescence after 18 h of levamisole exposure in adapting mutants, while WT worms showed no decrease of L-AChR synaptic fluorescence. We controlled the specificity of this reduction by quantifying other synaptic receptors present at the NMJ, i.e., the nicotine-sensitive homomeric AChR and the type A GABA receptor [6,21] (Figure 2c,d). The level of these receptors did not change in *lev-10* mutants upon 18 h exposure to levamisole.

The decrease in L-AChR at the synapse could be due to a general decrease in the receptor at the membrane or to their delocalization outside the synapse. To distinguish between these two hypotheses, we measured the currents induced by levamisole on open-cut preparations of *lev-10(0)* worms after 2 or 18 h of levamisole exposure (Figure 2e). After 18 h on levamisole, levamisole current amplitudes were much smaller as compared to those recorded after 2 h on levamisole, and were almost absent at −30 mV. We also measured the membrane potential of muscle cells from *lev-10(0)* mutants without levamisole or after 2 and 18 h of levamisole exposure. As previously described [21], muscle cells were depolarized after 2 h on levamisole (Figure 2f). However, after 18 h, membrane potential returned back to values similar to those before levamisole exposition (Figure 2f).

We concluded from these results that prolonged exposure to levamisole causes a reduction of levamisole-sensitive receptors at the plasma membrane of adapting mutants. We subsequently looked for pathways which could play a role in this adaptation mechanism of specific receptor deletion.

### 3.4. Calcineurin Is Necessary for Levamisole Adaptation

In order to unravel the pathways that could lead to levamisole adaptation, we tested a set of candidate genes. In the adapting *lev-10(0)* mutant context, we targeted genes that have been shown to play different roles related to calcium regulation and general cell functions. Among 16 tested genes, only *tax-6*, which encodes the active subunit of the calcium-dependent phosphatase calcineurin, seemed to be essential for adaptation (Table 1).

However, as *tax-6(0)* mutant worms exhibit a slow growth and a strong locomotion defect, the lack of adaptation in *lev-10(0); tax-6(0)* mutants could be due to a synthetic effect between general sickness of animals and levamisole toxicity. We thus used tissue-specific degradation of TAX-6/calcineurin in *lev-10(0)* mutants using the auxin-inducible degradation system [30]. Animals with ubiquitous degradation of TAX-6/calcineurin after the embryonic stage exhibit a slight reduction in body size but no gross alteration of locomotion, contrary to *tax-6(0)*. However, this ubiquitous degradation led to the suppression of levamisole adaptation in *lev-10(0)* mutants (Figure 3a). Degradation of TAX-6/calcineurin only in muscle cells (Appendix A) of *lev-10(0)* mutants also led to the suppression of adaptation, while degradation in other tissues had no impact (Figure 3a). We noticed a unique phenotype in *lev-10(0)* with degradation of muscle calcineurin: they first contracted in a similar manner to *lev-10(0)* but then stayed contracted and slowly hyper-contracted and shortened even more, instead of relaxing and progressively recovering mobility as *lev-10(0)* worms would (Figure 3b). Finally, we performed muscle degradation of TAX-6/calcineurin in *molo-1(0)* mutant and observed that adaptation was also suppressed (Table 1). In conclusion, the degradation of TAX-6/calcineurin in muscle tissue appeared to be both necessary and sufficient to suppress adaptation to levamisole.

**Figure 3 cells-12-02201-f003:**
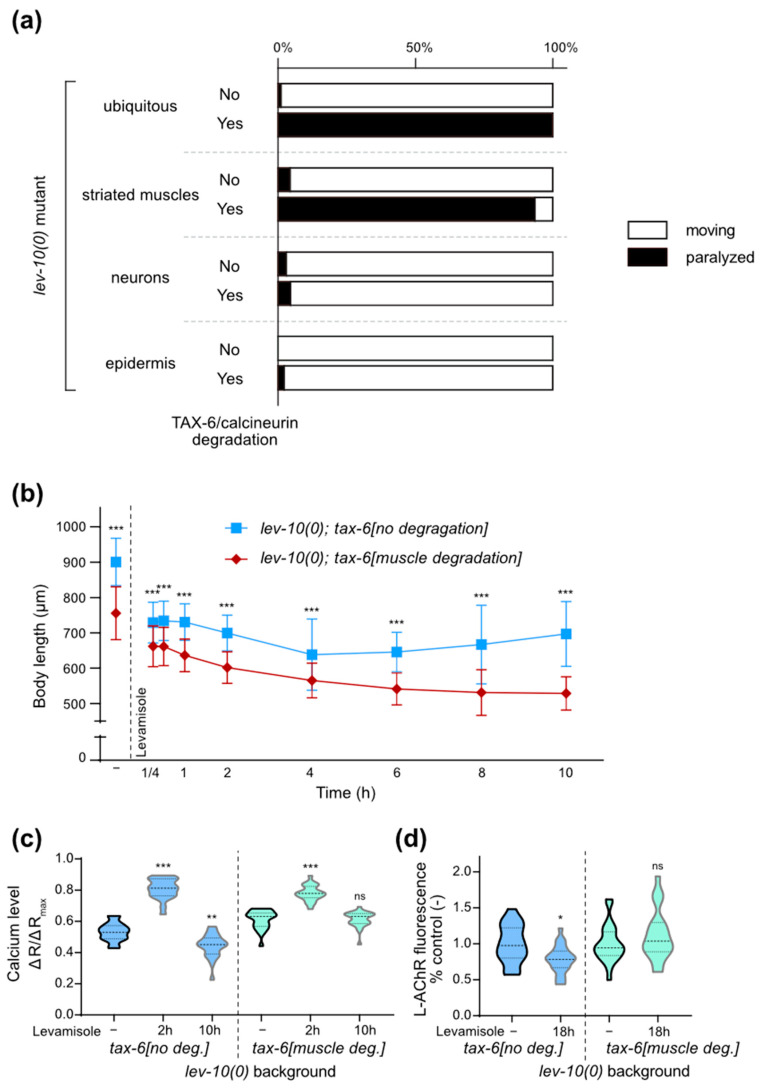
Muscle calcineurin is necessary for adaptation. (**a**) Paralysis of *lev-10(0)* with TAX-6/calcineurin tissue-specific degradation in different tissues after 18 h of 1 mM levamisole exposure, with or without 1 mM auxin applied from L4 stage. All animals carried *lev-10(kr26); tax-6(kr423[tax-6-aid-mNG])* and additionally either *Peft-3::TIR1-eBFP* (ubiquitous degradation, n = 277 without auxin, n = 189 with auxin), *Pmyo-3::TIR1-eBFP* (muscle degradation, n = 231 without auxin, n = 216 with auxin), *Prab-3::TIR1-eBFP* (neuron degradation, n = 194 without auxin, n = 173 with auxin), *Pdpy-7::TIR1-eBFP* (epidermis degradation, n = 168 without auxin, n = 181 with auxin). (**b**) Evolution of *lev-10(kr26)* length during levamisole exposure with or without muscle TAX-6/calcineurin degradation. *lev-10(kr26); tax-6(kr423[tax-6-aid-mNG]); Pmyo-3::TIR1-eBFP* animals were grown from the egg stage on plates without (no degradation) or with (muscle degradation) 1 mM auxin. Length was measured from pharynx to anus without levamisole treatment (-) (n = 122 without muscle TAX-6/calcineurin degradation, n = 126 with degradation) and after 15 min (n = 133 and n = 126 respectively), 30 min (n = 124 and n = 112, respectively), 1 h (n = 103 and n = 108, respectively), 2 h (n = 121 and n = 119, respectively), 4 h (n = 120 and n = 118, respectively), 6 h (n = 108 and n = 128, respectively), 8 h (n = 105 and n = 122, respectively) and 10 h (n = 104 and n = 140, respectively) of levamisole treatment. Kruskal–Wallis test *p* < 0.0001, Dunn’s post tests indicated on graph compare genotypes at equal time points. (**c**) Evolution of muscle calcium from *lev-10(kr26); tax-6(kr540[tax-6-aid-eBFP); Pmyo-3::TIR1-eBFP* during 1 mM levamisole exposure. Ratio between green and red muscle fluorescence were quantified from freely moving *lev-10(kr26); tax-6(kr540[tax-6-aid-eBFP); Pmyo-3::TIR1-eBFP* grown from the egg stage on plates without (no. deg.) or with (muscle deg.) 1 mM auxin without levamisole treatment (-) (n = 21 and n = 23, respectively), after 2 h (n = 22 and n = 21, respectively) or 10 h of levamisole exposure (n = 17 and n = 25, respectively). Welch ANOVA *p* < 0.0001; Dunnett’s T3 post tests indicated on the graph compare control condition to levamisole condition; *p* > 0.05 at 2 h with or without TAX-6/calcineurin degradation, *p* < 0.005 in control condition with or without TAX-6/calcineurin degradation, *p* < 0.0005 for 2 h/10 h without or with calcineurin degradation and at 10 h with or without TAX-6/calcineurin degradation. (**d**) Quantification of L-AChR content present at the dorsal nerve cord of *lev-10(kr26); tax-6(kr423[tax-6-aid-mNG]); Pmyo-3::TIR1-eBFP* grown from the egg stage on plates without (no. deg.) or with (muscle deg.) 1 mM auxin without levamisole treatment (n = 21 and n = 24, respectively) or after 18 h on 1 mM levamisole plates (n = 24 and n = 22, respectively). The fluorescence value of the dorsal cord was normalized to the mean value of the corresponding control condition. Mann–Whitney tests compare levamisole to control condition within a same genotype. For all *p* values: *** *p* < 0.0005, ** *p* < 0.005, * *p* < 0.05, ns (not significant) *p* ≥ 0.05.

**Table 1 cells-12-02201-t001:** Paralysis of animals exposed to 1 mM levamisole during 18 h. Absence of movement was assessed from one-day adults exposed to 1 mM levamisole, with or without 1 mM auxin applied from the egg stage, with or without *lev-10(kr26)* mutation and *Pmyo-3::TIR1-eBFP* transgene in genetic background. The most important strains are highlighted by grey and bold.

	Genotype	Auxin	Paralyzed	Moving	n
*lev-10(0)*	*aak-1(tm1944); aak-2(ok524)*	-	2%	98%	424
	*cmk-1(ok287)*	-	3%	97%	351
	*crh-1(n3315)*	-	7%	93%	482
	*daf-16(mu86)*	-	4%	96%	645
	*fzo-1(tm1133)*	-	1%	99%	164
	*gar-3(gk305)*	-	3%	97%	801
	*lgg-1(pp141)*	-	2%	98%	827
	*mca-3(ar493)*	-	4%	96%	274
	*mcu-1(ju1154)*	-	3%	97%	681
	*mef-2(gk633)*	-	2%	98%	585
	*slo-1(cx29); slo-2(nf100)*	-	2%	98%	863
	*tax-6(db60)*	-	**90%**	**10%**	192
	*unc-43(e408); zwIs102[Prab-3::yfp-unc-43]*	-	6%	94%	574
	*unc-68(r1162)*	-	1%	99%	457
*WT*	N2	-	**100%**	**0%**	797
	*lev-10(kr26)*	-	3%	97%	831
	*unc-29(e1072am)*	-	2%	98%	554
	*molo-1(kr100)*	-	1%	99%	257
	*molo-1(kr100); tax-6(kr423-aid-mNG); krSi55[Pmyo-3::TIR1-eBFP]*	-	6%	94%	395
	*molo-1(kr100); tax-6(kr423::degron::mNG); krSi55[Pmyo-3::TIR1-eBFP]*	1 mM	**75%**	**25%**	262
*lev-10(0); krSi55[Pmyo-3::TIR1-eBFP* *]*	*tax-6(kr423::aid::mNg)*	-	5%	95%	514
	*tax-6(kr423::aid::mNg)*	1 mM	**76%**	**24%**	381
	*tax-6(kr540::aid::BFP) IV*	-	4%	96%	545
	*tax-6(kr540::aid::BFP)*	1 mM	**80%**	**20%**	423
	*crh-1(bab180::aid)*	-	4%	96%	469
	*crh-1(bab180::aid)*	1 mM	1%	99%	212
	*crtc-1(kr450::aid)*	-	6%	94%	413
	*crtc-1(kr450::aid)*	1 mM	3%	97%	325

### 3.5. Calcineurin Is Involved in Calcium Level Restoration and Is Required for L-AChR Downregulation

Next, we analyzed muscle calcium concentration over prolonged exposure to levamisole in animals with or without degradation of muscle TAX-6/calcineurin. In the absence of levamisole, we observed a very slight increase in calcium concentration in *lev-10(0)* mutants when TAX-6 was degraded in muscles (Figure 3c). After 2 h of levamisole exposure, calcium reached the same level in *lev-10(0)* muscles with or without degradation of TAX-6. At 10 h, the calcium concentration decrease was less pronounced in *lev-10* mutants when TAX-6 was degraded in muscle (Figure 3c).

Altogether, our results suggested that muscle calcineurin is involved in the reduction of calcium level during adaptation. We hypothesized that this could be due to an effect of calcineurin on L-AChR content decrease during adaptation. We thus compared the evolution of synaptic L-AChR content after levamisole overnight exposure with, or without, the degradation of TAX-6 in muscle. In *lev-10(0)* background, without TAX-6/calcineurin degradation, synaptic L-AChR content was decreased by 22% (Figure 3d) after 18 h exposure to levamisole. Degradation of TAX-6/calcineurin in muscle fully suppressed the decrease in L-AChR at synapses. These results show that calcineurin is necessary for the degradation of synaptic L-AChR during long exposure to levamisole of adapting mutants.

### 3.6. Mitochondria Remodeling Parallels Adaptation to Levamisole

One would expect that the prolonged depolarization and calcium concentration increase observed in adapting strains could have stressful effects on muscle cells. We thus followed stress indicators to understand how muscle cells can cope with these stressful conditions. We first investigated the evolution of mitochondria morphology over the course of adaptation.

Using a single-copy reporter encoding the fluorescent wScarlet anchored to the mitochondrial outer membrane of muscle [32], we visualized the overall morphology of mitochondria after 2 h and 18 h of levamisole exposure (Figure 4a,b). After 2 h exposure to levamisole, mitochondria became fragmented both in WT and *lev-10(0)*. After 18 h on levamisole, mitochondria fragmentation had increased in WT muscle cells, while mitochondria from *lev-10(0)* were elongated and re-formed network.

In parallel, we measured cytosolic ATP concentration during levamisole exposure in animals by expressing the ratiometric Queen-2m probe in muscle cells [37,40] (Figure 4c). At 2 and 4 h of levamisole exposure, ATP concentration was lowered in both WT and *lev-10(0)* worms, compared to control conditions. In WT animals, ATP levels were still relatively low after 8 and 18 h of levamisole exposure, while they returned to initial levels after 8 h in *lev-10(0)* mutants.

ATP decrease might either reflect an impairment of ATP production or an increase in ATP consumption caused by increased activities of cellular pumps in response to abnormal voltage and ionic gradients. We tested whether mitochondria re-fusion was necessary for adaptation by monitoring locomotion on levamisole plates of *lev-10(0)* mutants carrying an additional mutation in *fzo-1*. FZO-1 is the ortholog of the mammalian mitofusin, an outer membrane GTPase required for mitochondrial fusion [41]. *lev-10(0); fzo-1(0)* mutants were able to adapt to levamisole overnight, in a similar manner to the single mutant *lev-10(0)* (Table 1). Thus, mitochondrial re-fusion occurs during adaptation but is not necessary for movement recovery.

### 3.7. Movement Recovery Is Associated with Increase in Autophagic Activity and Lysosomal Activity

We then investigated autophagy activity in muscle cells during levamisole exposure, since autophagy has been known to be stimulated as a protective mechanism against several forms of stress, including mitochondrial damage [42]. We used transgenic worms expressing in muscles a GFP-tagged version of LGG-1 [33], the *C. elegans* ortholog of GABARAP (Figure 5a). The number of autophagic vesicles per muscle cells increased in adapted *lev-10(0)* mutants on levamisole, while it did not in WT (Figure 5b). We verified the specificity of the particles counted as true autophagosomes by comparing with a strain in which a mutation on *lgg-1* prevents its aggregation to autophagosomes [33,34]: in this strain, particles would correspond to cytosolic aggregates and not autophagosomes. We observed very few particles in this control strain. The increase in number of autophagic vesicles in adapted worms was also observed using the mCherry-GFP-LGG-1 probe [35] that allowed the visualization of autophagosomes in green and red and autolysosomes in red only (Appendix A). Using this probe, we observed a strong tubulation of the autolysosomes after levamisole treatment (Appendix A), which is very peculiar.

To further investigate the lysosomal activity, we engineered a strain expressing a BFP-tagged version of LMP-1 in muscle. LMP-1 is the homolog of mammalian LAMP-1 which is located at the surface of lysosomes [43]. We confirmed the formation of lysosome network in *lev-10(0)* worms exposed to levamisole (Figure 5c,d). The lysosomes were longer for *lev-10(0)* worms after 18 h on levamisole, notably because of the presence of a small number of very long tubule networks (Appendix A).

Finally, we checked whether autophagy was required for levamisole adaptation using the *lgg-1(G116AG117 *)* [44] mutant, which shows no developmental defect but has no autophagic activity. The blocking of autophagy did not prevent adaptation of *lev-10(0)* mutants (Table 1).

### 3.8. Calcineurin Is Necessary for Mitochondria Shape Modifications but Not for Autophagic and Lysosomal Changes

We investigated the possible implication of muscle TAX-6/calcineurin in the homeostatic processes taking place during levamisole adaptation. Autophagic activity increased overnight on levamisole both with and without muscle TAX-6 (Figure 6a). Lysosomes tubulation increased overnight on levamisole both with and without TAX-6/calcineurin (Figure 6b). Tubulation seemed to be even more important for *lev-10(0)* mutants after TAX-6 degradation in muscle cells: more cells exhibited lysosomes that formed a complex network. In contrast, the re-fusion of mitochondria observed in *lev-10(0)* mutants during adaptation was suppressed when TAX-6 was degraded in muscle (Figure 6c). Thus, calcineurin plays a role in mitochondria remodeling during adaptation, but has no impact on autophagy and lysosome morphology.

## 4. Discussion

Levamisole adaptation in *C. elegans* was described more than 40 years ago; yet the cellular mechanisms underlying this phenomenon have never been studied. Here we showed that calcium plays a pivotal role in adaptation. Cytosolic calcium concentration first increases in striated muscle cells and remains high for several hours. During this phase, animals are paralyzed. In muscle cells, mitochondria are fragmented, cytosolic ATP concentration decreases and lysosomes start to form tubulated networks. The second phase occurs when animals resume locomotion. Cytosolic calcium is restored at low level. This is correlated with a decrease in L-AChR content at the muscle plasma membrane. At that time, mitochondria have recovered their initial morphology, cytosolic ATP level is restored, lysosome tubulation has increased and autophagy vesicles are visible in muscle cells. The decreases in calcium and L-AChR content are abolished in mutants defective for the calcium-dependent phosphatase calcineurin.

From our results, we propose the following model to explain how animals can adapt to levamisole. Levamisole opens L-AChRs, which do not desensitize [45], and this triggers depolarization of muscle cell membrane and the activation of voltage-gated calcium channels. These channels do not inactivate completely [21] and allow calcium flow into the cells, leading to an increase in cytosolic calcium concentration. This sustained calcium increase stimulates calcineurin activity through calmodulin binding. Decrease in L-AChR content is likely a consequence of calcineurin activation. Once L-AChR level has decreased, depolarization of muscle membrane stops and voltage-gated calcium channels are not activated anymore. Pumps and exchangers extrude calcium from the cytosol of muscle cells, and locomotion can resume, probably using levamisole-insensitive AChRs. This model is supported by the following observations: (i) calcium first increases then decreases during adaptation, (ii) degradation of calcineurin in muscle cells prevents the decrease in L-AChR content and in cytosolic calcium concentration, (iii) when the initial sustained calcium increase does not occur, as in animals that are unable to adapt to levamisole, L-AChR content stays stable over time, and (iv) membrane potential is depolarized when calcium is high and close to that observed without levamisole exposure when locomotion has resumed and calcium is low.

Our experiments give evidence that the general loss of levamisole sensitivity observed in adapting animals involves L-AChR downregulation. In most cases, nicotinic receptors have been described to undergo upregulation in response to chronic agonist exposure. A well-studied upregulation is the one observed for neuronal receptors in response to nicotine [46,47]. The effect of agonists on muscle nicotinic AChRs seems less clear: nicotine has been reported to upregulate several muscle subunits in cultured cells [48,49,50]; however, earlier studies have given evidence of a possible downregulation following either chronic exposure to nicotinic agonists, such as carbamylcholine in cultured cells, or in vivo injection of acetylcholine esterase inhibitors [51]. In *C. elegans*, non-lethal low dose of levamisole overnight has been shown to have a downregulating effect on vulval muscle AChRs [52]. This downregulation takes place within a similar time window as that observed for levamisole adaptation: in both cases more than 12 h was necessary, suggesting that downregulation could be mediated at the transcriptional level.

We showed that downregulation of AChR in striated muscle cells under levamisole exposure was conditional to the presence of the muscle calcium-dependent phosphatase calcineurin. Calcineurin has been reported to regulate AChR properties or dynamics in a few studies. First, calcineurin has been shown to modulate AChR desensitization in primary culture of mammalian chromaffin medullary cells [53] and of cortical neurons [54], and in isolated muscle fibers from snake [55]. Second, calcineurin has been reported to play a role in AChR dispersal and clustering in cultured *Xenopus* muscle cells and mouse myotubes: in this study the authors proposed that calcineurin promotes clustering either by acting on the clustering machinery or on the dystrophin-associated stabilizing complex [56]. Two other studies have given indirect evidence of a potential effect of calcineurin on AChR: (i) a calcineurin-dependent inhibition of ACh-induced exocytosis has been described in striatal synaptosomes, and it has been hypothesized that calcineurin’s target could be either the exocytosis machinery or the AChR themselves [57], and (ii) the activity-dependent rundown of the current flowing through alpha7-AChR observed in dissociated embryonic chick ciliary ganglia has been shown to increase when calcineurin was inhibited, suggesting that calcineurin plays a regulatory role on this AChR [58]. Finally, TAX-6/calcineurin was biochemically copurified with L-AChR subunits in *C. elegans*, suggesting a direct or indirect physical association between the two proteins [59]. However, the time scale of AChR downregulation during levamisole adaptation seems poorly in line with a direct action of calcineurin on L-AChRs. We favorize two hypotheses. First, calcineurin could act on a protein involved in L-AChR biosynthesis. In this case, dephosphorylation of this protein by calcineurin would inhibit its function and lead to a decrease in L-AChR biosynthesis. RIC-3 could be a good candidate since (i) this endoplasmic reticulum protein is required for AChR maturation [60], and (ii) calcineurin has been reported to modulate its activity [61]. The first hypothesis would imply that L-AChRs have a lifetime of a few hours. Half-life has been reported for N-AChR and seems to be pretty short [62]; however, it has not been determined for the L-AChR. Regarding the second hypothesis, the regulation of calcineurin on L-AChR would have a transcriptional origin. This transcriptional regulation could activate transcription of genes inhibiting L-AChR function or biosynthesis. Alternatively, the transcriptional regulation could promote L-AChR degradation or endocytosis. Chaya et al. have shown that decreasing L-AChR endocytosis leads to an increase in L-AChR content at the plasma membrane and to hypersensitivity to levamisole [63]. In the absence of NFAT in *C. elegans*, the best-known targets of calcineurin are the transcription factors DAF-16/FOXO [64] and CRH-1/CREB [65], and the transcriptional co-activator CRTC-1 [65]. However, we did not observe any defect in levamisole adaptation in animal mutants for any of these proteins (Table 1), suggesting that they are not involved in L-AChR downregulation.

Our study indicates that levamisole adaptation is accompanied by strong homeostatic mechanisms. In addition to L-AChR downregulation, we observed changes in mitochondria, lysosome and autophagic activities. Mitochondria are highly plastic and their morphology undergo dramatic changes [66]. During levamisole adaptation, mitochondria first go through fragmentation and then fused again. The change in mitochondria morphology could be a consequence of calcium homeostasis modification. Indeed, mitochondria dynamics have been already described to be influenced by free cytosolic calcium concentration in mammalian cultured cells [67], myocytes [68] and also in *C. elegans* striated muscle. In this organism, high calcium levels, correlated with mitochondria fragmentation, have been described in muscle cells of *sel-12*/presenilin mutants [69]. Mitochondria from *C. elegans* body-wall muscles have been shown to undergo fragmentation when subjected to calcium leak due to heat shocks [70]. The authors proposed that calcium leak triggers mitochondria fragmentation through a calcineurin pathway: sustained levels of calcium would activate calcineurin that would dephosphorylate the fission protein DRP-1 and promote its translocation to mitochondria. This calcineurin-dependent activation of mitochondria fission was initially reported in HeLa cells [71]. However, this pathway alone cannot account for the mitochondria fragmentation that we observed during the first hours of levamisole exposure. Indeed, mitochondria went through fragmentation even when calcineurin was depleted in muscle cells.

In our study, we showed that autophagy seemed activated in mutants with partially defective L-AChR, as *lev-10* mutants. Since autophagy has been shown to be regulated by changes in calcium homeostasis [72], we can hypothesize that the increase in intracellular calcium during the first hours of levamisole leads to a sustained activation of autophagy. A promoting role of autophagy on mitochondria fusion has been reported in *C. elegans* epidermis after acute heat stress [73]. However, such a mechanism does not seem to be involved in levamisole adaptation, at least in adapting mutants with TAX-6 calcineurin depleted in muscle cells: in these mutants, autophagy is increased; however, mitochondria are still fragmented. Activation of autophagy is not required for levamisole adaptation since (i) *lev-10* mutants with impaired autophagy can adapt to levamisole, and (ii) *lev-10* mutants with depletion of muscle TAX-6/calcineurin exhibit increased autophagy but cannot adapt to levamisole.

Finally, we observed lysosome tubulation in muscle cells from *lev-10* mutants exposed to levamisole. Tubulation appeared early and was not dependent on TAX-6/calcineurin. Lysosomal tubulation was first described in macrophages and was shown to be tightly linked to immune cell activation [74]. Lysosome tubulation has also been described as a mechanism to recycle phagolysosomes and autolysosomes after a major autophagic event [75]. Regarding levamisole adaptation, this does not seem to be the case since lysosome tubulation appears before the increase in autophagic activity. In *C. elegans*, lysosome tubulation has been described in epidermis during molting [76] and in intestine during starvation [77], two conditions that require high proteolytic activity. Tubulation has also been reported during aging in several tissues of *C. elegans* [78]. The functional consequences of lysosome tubulation are not well understood [74].

## 5. Conclusions

Animals that are able to adapt to levamisole initially undergo an increase in cytosolic calcium, lasting several hours. After this initial phase, L-AChR are downregulated through a calcineurin-dependent pathway, leading to a fall in cytosolic calcium and resumption of locomotion. During the adaptation process, major changes in cellular homeostasis are observed in mitochondria, lysosome and autophagy dynamics. Our study highlights a new experimental paradigm for studying the effects of a sustained cytosolic calcium increase on muscle cells. Increased cytosolic calcium concentration in muscle cells has been reported in pathological situations, such as malignant hyperthermia [79,80] or Duchenne’s dystrophy [81,82,83]. A better understanding of how cells can adapt to calcium-induced stress could contribute to the development of new therapeutic strategies to treat these pathologies.

## Figures and Tables

**Figure 1 cells-12-02201-f001:**
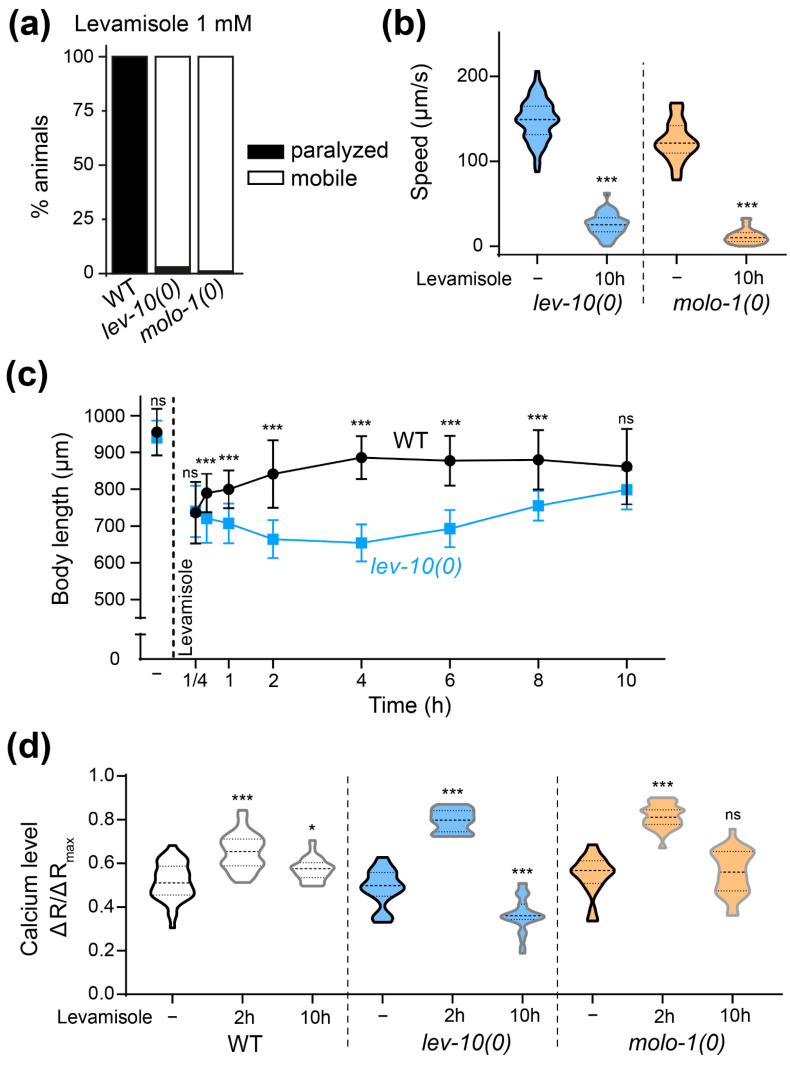
Locomotion recovery during levamisole adaptation is correlated with a decrease in muscle contraction and cytosolic calcium concentration. (**a**) Paralysis of WT (n = 797), *lev-10(kr26)* (n = 831) and *molo-1(kr100)* (n = 257) animals after 18 h of 1 mM levamisole exposure. Data set is the same as that in Table 1. (**b**) Mean speed of *lev-10(kr26)* and *molo-1(kr100)* animals placed for 10 h on control plates (n = 72 and n = 22, respectively) or 1 mM levamisole plates (n = 70 and n = 48, respectively). Speed was measured as mean centroid displacement for 50 s directly following mechanical stimulation. Kruskal–Wallis *p* < 0.0001, Dunn’s post tests indicated on graph compare control condition to levamisole condition for each genotype; *p* > 0.05 for *lev-10* control/*molo-1* control; *p* < 0.05 for *lev-10* levamisole/*molo-1* levamisole. (**c**) Evolution of WT and *lev-10(kr26)* length during 1 mM levamisole exposure. Length was measured from pharynx to anus of WT and *lev-10* animals without levamisole treatment (-) (n = 115) or after 15 min (n = 129 and n = 132, respectively), 30 min (n = 121 and n = 114, respectively), 1 h (n = 108 and n = 111, respectively), 2 h (n = 113 and n = 116, respectively), 4 h (n = 101 and n = 103, respectively), 6 h (n = 89 and n = 102, respectively), 8 h (n = 83 and n = 117, respectively) and 10 h (n = 68 and n = 123, respectively) of levamisole treatment. Kruskal–Wallis test *p* < 0.0001, Dunn’s post tests indicated on graph compare genotypes at equal time points. (**d**) Evolution of muscle calcium during 1 mM levamisole exposure. Ratio between green and red muscle fluorescence were quantified from freely moving WT, *lev-10(kr26)* and *molo-1(kr100)* animals expressing *Pmyo-3::GCaMP-6-mCherry*, before levamisole (n = 44, n = 21 and n = 25, respectively), after 2 h (n = 36, n = 14 and n = 27, respectively) or 10 h of levamisole exposure (n = 43, n = 29 and n = 27, respectively). Welch ANOVA *p* < 0.0001; Dunnett’s T3 post tests indicated on the graph compare control condition to levamisole condition; *p* > 0.05 for all tests between WT, *lev-10* and *molo-1* before levamisole exposure, for *lev-10*/*molo-1* at 2 h, for WT/*molo-1* at 10 h; *p* < 0.005 for WT/*lev-10* and WT/*molo-1* at 2 h, for WT/*lev-10* and *lev-10*/*molo-1* at 10 h, for WT 2 h/10 h, for *lev-10* 2 h/10 h and for *molo-1* 2 h/10 h. For all *p* values: *** *p* < 0.0005, * *p* < 0.05, ns (not significant) *p* ≥ 0.05.

**Figure 2 cells-12-02201-f002:**
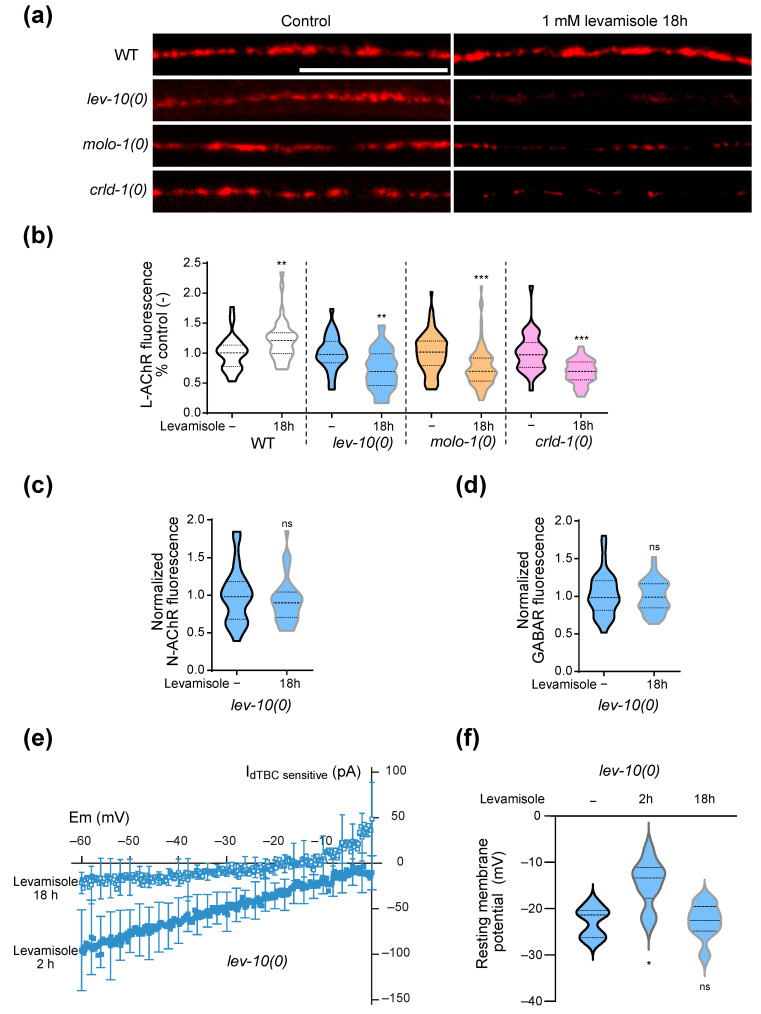
L-AChR are downregulated at NMJs during adaptation. (**a**) Representative confocal pictures of fluorescent L-AChR present at the dorsal nerve cord of WT, *lev-10(kr26)*, *molo-1(kr100)* and *crld-1(kr407)* animals carrying *unc-29(kr208::TagRFP-T).* Images are displayed with head side oriented left and summed in depth. Images look-up-tables are fixed for each genotype but different between genotypes for better visualization. For all panels, worms were exposed to 1 mM levamisole on standard growth plates for 18 h; control condition corresponded to 18 h on standard growth plates. Scale bar: 20 μm. (**b**) Quantification of L-AChR content present at the dorsal nerve cord of WT, *lev-10(kr26)*, *molo-1(kr100)*, *crld-1(kr407)* carrying *unc-29(kr208::TagRFP-T)* without levamisole treatment (n = 34, n = 33, n = 82 and n = 33, respectively) or after 18 h on 1 mM levamisole plates (n = 38, n = 43, n = 87 and n = 36, respectively). The fluorescence value of the dorsal cord was normalized to the mean value of the corresponding control condition. Multiple Mann–Whitney tests compare levamisole to control condition within a same genotype. (**c**) Quantification of synaptic nicotine-sensitive AChR (N-AChR) from dorsal nerve cord of *lev-10(kr26)* mutants carrying *acr-16(kr440::wScarlet)* without levamisole treatment (n = 33) or after 18 h on 1 mM levamisole plates (n = 38). The fluorescence value of the dorsal cord after levamisole treatment was normalized to the mean value of the corresponding control condition. Mann–Whitney tests *p* = 0.3541. (**d**) Quantification of synaptic GABA receptor (GABAR) from dorsal nerve cord of *lev-10(kr26)* mutants carrying *unc-49(kr296::RFP)* without levamisole treatment (n = 33) or after 18 h on 1 mM levamisole plates (n = 38). The fluorescence value of the dorsal cord after levamisole treatment was normalized to the mean value of the corresponding control condition. Mann–Whitney tests *p* = 0.7614. (**e**) dTBC-sensitive currents of *lev-10(kr26)* muscle cells after 1 mM levamisole exposure during 2 h (n = 10) or 18 h (n = 9). Currents were recorded in the presence of 0.01 mM levamisole, with and without 0.1 mM dTBC to extract dTBC-sensitive currents. Mean currents were plotted. Standard deviations were plotted every 10 points. Mann–Whitney tests; *p* < 0.05 at −10 mV, *p* < 0.005 at -50, -20 and 0 mV, *p* < 0.0005 at −60, −40, −30 mV. (**f**) Membrane resting potentials of *lev-10(kr26)* muscle cells without levamisole treatment (n = 7) or after 2 (n = 14) or 18 h (n = 9) on 1 mM levamisole. Kruskal–Wallis *p* = 0.0012, Dunn’s post tests indicated on graph tests compare levamisole and control condition; *p* < 0.05 for 2 h/18 h of levamisole. For all *p* values: *** *p* < 0.0005, ** *p* < 0.005, ns (not significant) *p* ≥ 0.05.

**Figure 4 cells-12-02201-f004:**
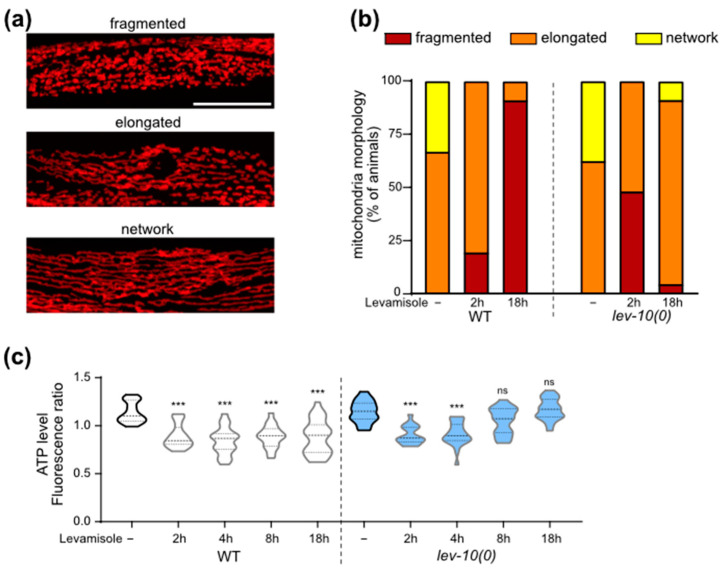
Levamisole induces muscle mitochondrial fragmentation that disappears during adaptation. (**a**) Representative confocal pictures of muscle mitochondrial morphology. Mitochondria were visualized in body-wall muscle cells from worms expressing a single copy of *Pmyo3::tom-20N-wScarlet*. Look-up-table is independent for each image and displays maximum pixel value in depth. Scale bar: 20 μm. (**b**) Evolution of muscle mitochondria fragmentation during adaptation. Mitochondria morphology was assayed in WT (n = 21) and *lev-10(kr26)* (n = 21) animals expressing *Pmyo3::tom-20N-wScarlet*. Worms were exposed to 1 mM levamisole for 18 h. For each worm, a minimum of 7 muscle cells between terminal bulb and anus were evaluated according to scale described in A. The most attributed grade for each worm was retained. (**c**) Quantification of muscle ATP during levamisole exposure. WT and *lev-10(kr26)* animals expressing *Pmyo-3::Queen-2m* were placed on control (-) (n = 18 for both genotypes) or 1 mM levamisole plates for 2 h (n = 19 and n = 16, respectively), 4 h (n = 23 and n = 24, respectively), 8 h (n = 18 for both) or 18 h (n = 28 and n = 21, respectively). Samples were excited at 405 nm and 488 nm and fluorescence ratio was plotted for each animal as an average of the three areas. Welch ANOVA test *p* < 0.0001; Dunnett’s T3 post tests; *p* values indicated on the graph compare levamisole to control condition of the same genotype; *p* > 0.05 for WT control/*lev-10* control, WT 2 h levamisole/*lev-10* 2 h levamisole, WT 4 h levamisole/*lev-10* 4 h levamisole; *p* < 0.05 for WT 8 h levamisole/*lev-10* 8 h levamisole; *p* < 0.0005 for WT 18 h levamisole/*lev-10* 18 h. For all *p* values: *** *p* < 0.0005, ns (not significant) *p* ≥ 0.05.

**Figure 5 cells-12-02201-f005:**
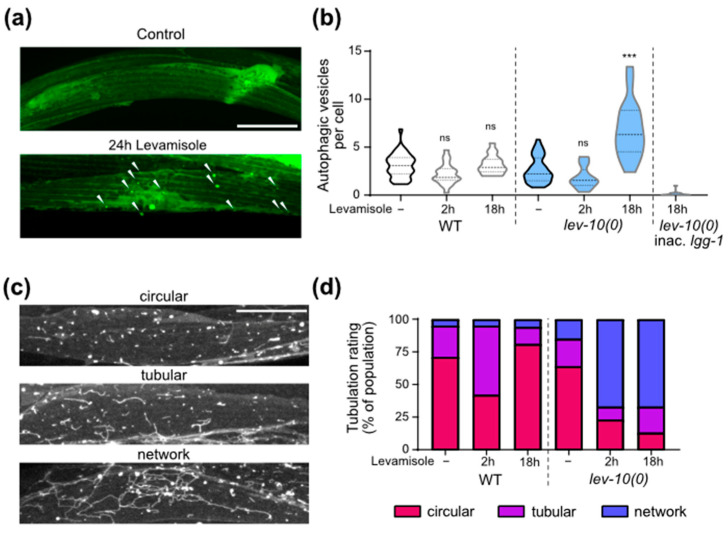
Muscle autophagic and lysosomal activities increase during adaptation. (**a**) Representative confocal images of muscle autophagosomes. Animals expressed *Pdyc-1::GFP-lgg-1*. Look-up-table is independent for each image. Maximum pixel value in depth was displayed. White arrowheads point to autophagosomes. Scale bar: 20 μm. (**b**) Quantification of muscle autophagic vesicles after levamisole exposure. Muscle autophagosomes were counted in WT and *lev-10(kr26)* animals expressing *Pdyc-1::GFP-lgg-1* without levamisole treatment (n = 35 and n = 37, respectively) and after 2 h (n = 21 and n = 22, respectively) or 18 h of levamisole exposure (n = 17 and n = 18, respectively). Vesicles were also counted from WT animals expressing *Pdyc-1::GFP-lgg-1G116A* (inac. *lgg-1*) to control for false positives (n = 15). Number of vesicles was determined in a minimum of 7 cells per worm and averaged by worm. Kruskal–Wallis *p* < 0.0001; Dunn’s post tests; *p* values indicated on the graph compare levamisole to control condition of the same genotype; *p* > 0.05 for WT/*lev-10* in control conditions and after 2 h of levamisole; *p* < 0.05 for WT/*lev-10*; *p* < 0.0001 for WT/inac. *lgg-1* after 18 h of levamisole. (**c**) Representative confocal images of lysosome morphology. Muscle lysosomes were visualized in worms expressing *Pmyo-3::lmp-1-eBFP*. Look-up-table is independent for each image and displays maximum pixel value in depth. Scale bar: 20 μm. (**d**) Evolution of muscle lysosomal morphology during levamisole exposure. Lysosome morphology was assessed in muscle cells of WT (n = 55 in control conditions, n = 45 and n = 16 after 2 or 18 h on levamisole) and *lev-10(kr26)* (n = 53 in control conditions, n = 48 and n = 15 after 2 or 18 h on levamisole) animals expressing *Pmyo-3::lmp-1-eBFP*. Lysosome tubulation was evaluated using the scale defined in c. Each worm was assessed with an image taken between the nerve ring and the vulva. For all *p* values: *** *p* < 0.0005, ns (not significant) *p* ≥ 0.05.

**Figure 6 cells-12-02201-f006:**
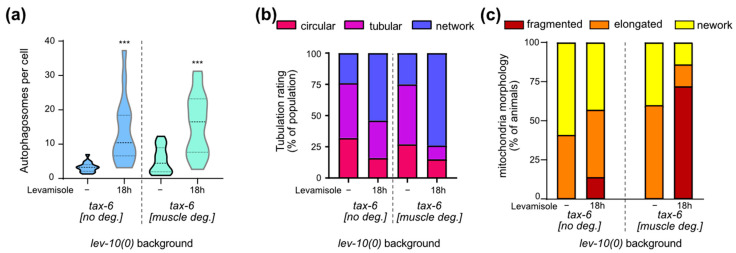
Muscle degradation of TAX-6/calcineurin has an impact on mitochondrial homeostasis during adaptation but not on autophagy or lysosome tubulation. (**a**) Quantification of muscle autophagosomes counted in *lev-10(kr26); tax-6(kr540[tax-6-aid-eBFP]); krSi55[Pmyo-3::TIR1-eBFP]; kagIs1[Pdyc-1::GFP-lgg-1]* animals expressing *Pdyc-1::GFP-lgg-1* without or with TAX-6/calcineurin degradation before levamisole treatment (n = 17 and n = 16, respectively) or after 18 h of levamisole exposure (n = 16 for both). Number of vesicles was determined in a minimum of 7 cells per worm and averaged by worm. Kruskal–Wallis *p* < 0.0001; Dunn’s post tests; *p* values indicated on the graph compare levamisole to control condition of the same genotype; *p* > 0.05 for control without TAX-6/calcineurin degradation/control with degradation and for 18 h without TAX-6/calcineurin degradation/18 h with degradation. (**b**) Evaluation of muscle lysosomal morphology after levamisole exposure from muscle cells of WT (n ≥ 15) and *lev-10(kr26)* (n ≥ 13) worms expressing *Pmyo-3::lmp-1-eBFP*. Lysosome tubulation was evaluated using the scale defined in Figure 5c. Each worm was assessed with an image taken between the nerve ring and the vulva. (**c**) Evaluation of muscle mitochondria morphology during adaptation. Mitochondria fragmentation was evaluated from body-wall muscle cells of *lev-10(kr26)* (n = 21) and WT (n = 21) worms expressing *Pmyo3::tom-20N::wScarlet*. Worms were exposed to 1 mM levamisole for 18 h. For each worm, a minimum of 7 muscle cells between pharyngeal terminal bulb and anus were evaluated according to scale described in Figure 4a. The most attributed grade for each worm was plotted. For all *p* values: *** *p* < 0.0005.

## Data Availability

The data that support the findings of this study are available from the corresponding author upon reasonable request.

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
