# Peer review of "Calcineurin-Dependent Homeostatic Response of C. elegans Muscle Cells upon Prolonged Activation of Acetylcholine Receptors"

_cells, 2023, doi:10.3390/cells12172201_

Round 1

Reviewer 1 Report

This work builds on a recent set of observations that reveal interesting underpinnings that underlie genetic determinants that reveal mechanism underling C.elegans adaptive response to protracted exposure the anthelmintic levamisole. The current submissions provides a molecular and cell biological investigations that suggest apparent Ca2+ signalling in muscle is responsible for the ability of mutants to express  a surprising movement after an initial paralysis evoked by whole organism exposure to levamisole. The key observation is that tax-6 calcineurin is important in expressing this mutant phenotype. I think the discussion reflects this central point well and relies on the time course of effects suggest that tax-6 acts distally from the receptor. There are some secondary observations regarding mitochondria and protein degradation that showcase some nice tools but seem unrelated to the central tenet of the paper. However, the value in these observations is in the way they inform about what is not important. The work makes for a useful complement to recent studies highlighting how levamisole and other forms cholinergic hyper stimulation execute behavioural plasticity.

Major points.

The authors offer that 1mM levamisole paralyses kills worms after several hours (ln 238-240). Given the time-course of the observations up to 18 hours, what is meant by “hours”?  And how is the worms death scored in the experiments that showed (citation) or were observed (data not shown). Some reference to this might help reader understand why 10 and 18 hours are variously and differently used in the experiments that map out the adaptive behaviour in the wild type versus the adapting mutants.

Figure 2 I think the data from the Unc-29 and acr-16 tagged receptors is a very important quality control for the selectivity of the effect. Representative images but certainly the quantification in suppl figure 2 B would be an excellent complement to data in figure 2 and should be shown in the main body of the text.

Figure 2 C is a complicated experiment. The reader might need to know that the filleted preparation removes the bath applied levamisole that has accumulated within the worm over an 18 hour experiment. Is there a reason why the control data from N2 worms is not shown (are they dead). It seems that labelling is wrong for 2 and 18 hrs.

Figure 2 D is there a reason why the N2 control is not shown.

Figure 3 have these experiments been performed in a wild-type background. This would be particularly interesting in the context of the ubiquitous and the wild type background given the pivot played by tax-6 in the model proposed in the discussion. Although not essential what is the outcome of 1mM dosing of the tax-6 mutant in a wild-type background?

Figure 4, 5 and 6 are driven by criteria driven observations made by experimenters blind to genotype. The supplementary might be used to highlight what the basis for criteria might be.

Minor points.

The allele name against genes is often denoted as (0). I have consulted those wiser than I and they suggest this best be (null).

Ln 32-33 Simplify? To “Levamisole selectively activates worms body wall muscle L-type receptor.”

Ln 43-44 Do you really want to say that levamisole only activates only one type of AChR. One might caution, caution.

Ln 46 remove totally ??

Ln47-48 confusing sentence and mutants would not exist at neuromuscular junction. Do you mean mutant receptors or functional receptors containing mutant receptors?

Ln 252 The pharyngeal data is interesting as it is now clear that the pumping response is a surrogate of L-type receptor activity (Gonzalez et al 2023). Just as the authors show the elevated pump-rate in face sustained drug exposure actually reflects an adaptation at the body wall muscle.

Ln 304 implies an interesting observation can the authors bin the data from hypercontracted and normal worms to show the association with Ca2+.

Figure 2B is the receptor fluorescence increased after 18 hours Levamisole in the wild type background.

Table 1 is useful but might be better organized to support the text. I suspect showing the mutant screen first and then progressing to section WT and finally the lev 10 (0);krSi55[Pmyo-3::TIR1-BFP].

Is there documentation of the response of the mutants (especially tax-6 db60) in a wild type background?

Figure 3 why is there a mismatch in the time window between what is shown in 3b (10 hours) and what is recorded in figure 3C and 3D.

Ln 427 use body wall muscle rather than muscular?

Figure 4. Remarkable fragmentation of the WT network after 18 hours. Is this recoverable.

Ln 470 instead of ATP levels were still lo; based on the data might it be better expressed as, relatively low??

534 clarify what is meant by Tubulation seemed to be even more important for lev-10 (0) after TAX-6 degradation.

589 ………Agonist-induced down-regulation is a common phenomenon except for nicotinic receptors. This is a bit unspecific. Desensitization would down regulate. I think citation against this would help reader or a more considered statement to clarify the point that is being made.

592 instead of “The most well-known upregulation is the one…..” Change to a well-studied??

615 a7=alpha7??

640 Is it worth defining the homeostatic response as complex involving shifting Ach sensitivity and other cellular phenomenon which do not contribute to movement recovery?

If there is enthusiasm for a cartoon to show diagrammatically the elements of the model the observations highlight.  The reader might appreciate this.

See suggested minor comments

Reviewer 2 Report

Manuscript Number: cells-2561366

Calcineurin-dependent homeostatic response of C. elegans muscle cells upon hyperactivation of acetylcholine receptors by Florin et al.

In this manuscript, the authors proved that calcineurin is involed in the homeostatic response of C. elegans muscle cells by regulating acetylcholine receptors after levamisole exposure, accompanied by remodelling mitochondria, autophagosomes and lysosomes. It is an very intresting work, and the manuscript needs a minor revision before it could be accepted for publication in this journal.

Detailed comments and suggestions:

1. Title: the authors should considered an alternative tiltle to extract their research, eg: Calcineurin-dependent homeostatic response of C. elegans muscle cells upon regulation of acetylcholine receptors, because "hyperactivation" might need more data to support it, the authors just find that the acetylcholine receptors decrease after  levamisole exposure.  

2. Abtsract: the importance of the present work should be addressed in more details.

3. Materials and Methods: page3 line98, finale is final? page5,line 228, d-TBC is dTBC?

4.Mitochondrial function only is indicated by ATP content, Oxygen Consumption Rate(OCR),mitochondrial membrane potential (MMP)etc. is recommand to be measured.

5. page12-13,line 404, 414, 422, legend of Figure3, no, deg. and muscle deg., and page15line505, legend of Figure5b, inac.lgg-1 means inactive lgg-1, those words suchlike should be deleted.

In  Materials and Methods section: page3 line98, finale is final? page5,line 228, d-TBC is dTBC?
